# Wearable Technology: A Wellbeing Option for Serving Police Officers and Staff? A Comparison of Results of a Pilot Study with Firearms Officers and a Group of Mixed Officers and Staff

**DOI:** 10.3390/ijerph21020186

**Published:** 2024-02-06

**Authors:** Carol Cox, James Yates, Freya O’Brien, Laura Pajon, Amanda Farrell, Moya Ward, Philippa McCabe, Adrian James, Isabella McNamara-Catalano

**Affiliations:** Liverpool Centre for Advanced Policing Studies, Liverpool John Moores University, Liverpool L3 5UX, UK; j.yates@2023.ljmu.ac.uk (J.Y.); f.obrien@ljmu.ac.uk (F.O.); l.pajonmoreno@ljmu.ac.uk (L.P.); a.l.farrell@ljmu.ac.uk (A.F.); m.o.ward@ljmu.ac.uk (M.W.); p.g.mccabe@ljmu.ac.uk (P.M.); a.d.james@ljmu.ac.uk (A.J.); i.m.mcnamaracatalano@ljmu.ac.uk (I.M.-C.)

**Keywords:** wearable technology (WT), police officers, wellbeing, firearms officers

## Abstract

The high-stress nature of policing contributes to deterioration of officer health and wellbeing as well as high levels of absenteeism and attrition. Wearable technology (WT) has been identified as a potential tool that can help in improving officer health and wellbeing. This pilot study aimed to give initial insight into acceptability and engagement with WT amongst officers. The study also aimed to uncover any notable areas for exploration in future research within the domain of officer health and wellbeing. Two groups were observed, firearms officers and a mixed group of officers. Participants wore the WT for an extended period, completed a variety of health and wellbeing questionnaires and discussed their experience in focus groups. Firearms officers and mixed group officers displayed similar sleep efficiency, but firearms officers have worse sleep consistency and sleep performance. Firearms officers appear to have higher HRV and a slightly lower resting heart rate. Both groups display reasonable acceptance of the use of WT, speaking favorably during the focus groups of how monitoring the data had improved their quality of life in terms of their understanding of sleep, wellbeing and how they had consequently completed lifestyle modification. WT offers some promise in managing officer health and wellbeing; studies with larger sample sizes are needed to confirm this.

## 1. Introduction

It is recognized that policing is a high-stress occupation [1], with pressures coming from both operational (e.g., witnessing traumatic events) and organizational factors (e.g., workload) [2,3,4,5]. The frequency and severity of trauma experienced by police officers place them at a higher risk of developing mental health problems [6]. Wellbeing issues may ensue for firearms officers due to prolonged uncertainty as they face investigation following carrying out elements of their role [7]. Research indicates that serving officers are often reluctant to join firearms teams as there is very little support for them if they deploy their weapon, many believing that the ongoing investigation after a shooting could involve the job dropping you “like a stone” [8]. Cox et al. (2023, p. 46) found that “the potential life changing consequences and stress if you were to shoot someone is a big factor in putting people off (Male Authorised Firearms Officer) (joining the firearms teams)” and “There is far too little support of firearms officers following a shooting… that is the sole reason I would never join a firearms unit (Female Non Authorised Firearms Officer) [8]. In a time where crime is becoming more complex, harder to solve, and higher harm outcomes are being seen [9], there is decreasing public trust in the police [10], making the issue of officer wellbeing more pertinent than ever.

Officers and police staff allocated in certain units/roles may be more prone to wellbeing issues than others. Firearms officers are tactical personnel who are deployed to high-risk, unpredictable incidents and have completed supplementary necessary specialist training in the use of firearms for such events, and complete additional physical training to deal with the greater demands of their work compared with regular officers [11]. The nature of their work includes potential exposure to violence, which can contribute to an elevated risk of psychological distress [12]. There is a paucity of research on the general well-being of firearms officers despite previous studies showing that, although rare, officer-involved shootings can be impactful events for their wellbeing [13]. The responsibility to make split-second decisions about using deadly force weighs heavily on them, with potential consequences that last long after the incident [13]. Investigations and scrutiny from departmental officials follow these incidents, and officers may face disciplinary action or changes in their assignments [13]. Legal actions in criminal and civil courts can add further stress and potential repercussions [13].

Officers may also experience media attention and harassment, contributing to psychological distress [14,15,16]. The debriefing process following a critical incident has potentially yielded adverse outcomes, including the development of PTSD [17]. PTSD symptoms experienced post-shooting can vary but often include physical issues, sleep disturbances, intrusive thoughts, anxiety, depression, anger, irritability, and substance abuse [13]. Officers may oscillate between being overly protective of their families and withdrawing from sources of support. Furthermore, these events are likely to impact the personal and professional networks surrounding the officer(s) [18,19].

Between 2020 and 2021, over 13,000 police officers in the UK were ‘signed off’ from work due to depression, anxiety, and PTSD [20]. Demands on officers have been compounded by previous staffing cuts due to austerity measures [21], the United Kingdom (UK) government, responded in 2019, by launching plans to recruit 20,000 new police officers in England and Wales by March 2023 (the ‘Uplift Programme’) [22]. Although this drive has increased the number of new officers, record numbers are leaving the profession [23], leaving gaps in capability and experience.

In recent years, the College of Policing (CoP), the professional body for the police in England and Wales, has prioritized wellbeing, subsequent support, and awareness for officers and police staff by establishing Oscar Kilo, the National Police Wellbeing Service, in 2016 and the Police Covenant in 2022. Oscar Kilo has created a priority workstream to take forward the sleep, fatigue, and recovery of police officers. This priority was based upon findings of the importance of sleep, fatigue, and the link to shift work from several surveys conducted by different sleep centers across the world [24,25,26]. These findings identified an acute problem within operational policing roles, particularly where shift working is required. As part of the ‘digital health revolution,’ a way in which both researchers and clinicians have sought to measure and understand sleep health is via the use of wearable technology (WT) [27].

WT encompasses a wide range of devices designed to be worn on the body, including accessories, clothing, and medical devices. WT includes smartwatches, fitness trackers, VR headsets, and smart jewelry. These devices typically incorporate internet connectivity, allowing data to be synced with other devices for analysis and tracking [28]. As well as being used to monitor and understand sleep, WT has been found to have wellbeing benefits in the general population, such as a reduction in sedentary time and improvements in mood [29].

It has also been deployed in various occupational sectors with the intention of replicating the wellbeing benefits seen in the general population; WT has the potential to improve work efficiency and physical wellbeing in employees, as well as reduce work-related injuries [30]. In the hotel/hospitality sector, increases in physical activity, healthy food consumption, and reduction in overall calorie intake have been observed when employees use WT, alongside improvements in staff engagement, job satisfaction, and organizational commitment [31]. A study of pharmaceutical employees using WT also found an increase in sleep duration as individuals progressed on the intervention [32].

Reviews and research on the use of WT have highlighted both benefits of and challenges to using WT for health improvements. Issues that may outweigh the positive uses of these devices include users’ perception and acceptance of wearing technology, as well as issues around privacy (e.g., data protection) and ethics (e.g., right to liberty) [33]. Examining WT use in employees in workplaces similarly has raised concerns surrounding the protection of data and confidentiality [34]. This may contribute to resistance to using WT and reduced acceptance of devices as a viable tool in preserving health and wellbeing. Additional challenges for integrating these devices into police officer standard practice could also be the renowned police sub-culture where a heightened vigilance and skepticism towards non-police bodies exists [35].

In the context of enhancing the wellbeing and performance of firearms officers in the UK who have experienced shooting incidents, WT interventions offer promising solutions. Various wearable devices, such as heart rate monitors, electrodermal activity sensors, and sleep trackers, can continuously monitor officers’ physiological responses. These devices enable timely intervention and support by detecting signs of stress, anxiety, or other emotional states.

The aim of this study was to establish if police officers, including those working in firearms roles, would engage with WT to support their wellbeing, and if the health data collected could identify any trends that could inform future wellbeing interventions and initiatives.

## 2. Materials and Methods

### 2.1. Study Overview

The study was approved by the Institutional Review Board (or Ethics Committee) of Liverpool John Moores University (ethics no 23/LCP/006). The study focused on the experience and perceived wellbeing of firearms police officers wearing the “WHOOP” band, comparing their results with a mixed group of police officers and police staff. Participants provided informed consent before data collection. Participants were provided with a “WHOOP” device and a free subscription for an initial six-month period. The “WHOOP” (Model 4.0, Manufactured: Boston, MA, USA) band is a wrist-worn device, which enables the user to extract information such as heart rate variability (HRV), resting heart rate (RHR) and respiration rate.

Prior to the participants’ use of the “WHOOP” band, officers completed an anonymized survey to track and measure health-related concerns such as anxiety, stress, depression, sleep or sleep quality, exercise, and alcohol consumption. Upon completion of the six months wearing the “WHOOP” bands, staff from both groups were asked to complete the same survey as a post-measure to identify changes in participants’ wellbeing or behaviors. Next, two focus groups were conducted with firearms officers and officers from the mixed police staff group, examining perceptions of the use of the wearable tech and related outcomes/changes in their wellbeing and behaviors.

Resources related to stress and trauma (such as hotlines, support groups, etc.) were provided at the end of the pre- and post-intervention surveys. Additional support was provided to all officers involved in the study and the Occupational Health Unit attended the on boarding with all firearms officers. The firearms officers have continual assessments by the Occupational Health Unit as standard, so any issues are raised and dealt with internally, this is not discussed with the research team for confidential reasons.

### 2.2. Participants

A convenience sampling strategy was used to examine and compare wellbeing data from firearms police officers (*n* = 12) and a group of police staff from various roles (*n* = 28). The firearms officers had satisfied selection, training and accreditation and were authorised by a Chief Officer to carry a firearm operationally. The small sample size was used as this was an initial pilot scheme to gauge feasibility of conducting more in-depth research in due course. All participants were officers from the same territory police organization in England. Table 1 provides a description of both participant groups.

### 2.3. Instruments

#### 2.3.1. Wellbeing Questions

Participants completed an anonymized pre- and post-intervention survey. The survey included four validated scales (described below) and self-perceived general wellbeing questions. These wellbeing questions were all self-reported and subjective to the individual to interpret their appropriate response. The wellbeing survey included questions on had the participants received training on stress and trauma, stress/anxiety, use of self-care, use of mindfulness, physical activity levels, getting enough sleep, and work–life balance.

#### 2.3.2. Depression Anxiety Stress Scales (DASS)

The DASS [36] has three nested subscales within the assessment. The subscales clinically examine diagnostic features of three mental health conditions: depression, anxiety, and stress.

#### 2.3.3. The Perceived Stress Scale (PSS)

The PSS [37] was used to examine officers’ perceptions of being under stress. Scores below 13 were considered to be low stress and scores 14–26 were considered to be moderate stress, with scores of 27+ being considered high stress.

#### 2.3.4. The Pittsburg Sleep Quality Index (PSQI)

The PSQI [38] organizes the responses into seven components for analysis, which feed into a global score for the whole assessment: subjective sleep quality (1 item), sleep latency (2 items), sleep duration (1 item), sleep efficiency (3 items), sleep disturbance (9 items), use of sleep medication (1 item), and daytime dysfunction (2 items). For this study, the authors modified the assessment and categories slightly, including demographic variables and variables related to shift work, age, gender, ethnic identity, and home situation (children in the home, relationship status, etc.).

#### 2.3.5. The Alcohol Use Disorders Identification Test (AUDIT-C)

The AUDIT-C [39] is a commonly used screening for problem drinking based on consumption of alcohol responses. Typically, the first three questions are administered, and if the respondent scores less than five, they indicate lower-risk drinking. Scores of five or higher indicate higher-risk drinking behaviors. In clinical assessment settings, they would then need to do the remaining seven questions as a part of a substance use disorder screening (SUDS). The authors added a moderate drinking risk category to this traditional assessment (scores 5–6), to allow for a more clearly distinguished outlier/higher-risk consumption behavior.

### 2.4. WT

“WHOOP” fitness bands demonstrate good validity and reliability [40,41] and provides a HRV measure which has been described as providing useful indication of the psychophysiological responses during specific occupational tasks and the subsequent recovery from those tasks in populations such as firearms officers [42]. The “WHOOP” bands were used to extract biometric data of the participants in the study. Data was anonymously provided by “WHOOP” to the research team. Upon completion of the six-month period wearing the band, the daily averages of participants on that given day were taken.

Anonymized biometric data was extracted and categorized into ten variables: average sleep minutes, sleep performance (i.e., time in bed sleeping divided by the sleep needed), sleep consistency (i.e., similarity between sleep and wake-up times), sleep efficiency (i.e., the percentage of time in bed actually sleeping), HRV (i.e., variance in time between heartbeats, known as R-R interval and measured in milliseconds, “WHOOP” calculates the root mean square of successive differences (RMSSD) between heartbeats, with a greater variability indicating a higher level of readiness to execute at a high level), workout (i.e., calculated as a binary variable of yes or no), RHR (i.e., time when the body is at its most restful state), average strain (i.e., average amount of physical and mental strain the body is under (a measure incorporating cardiovascular and muscular load)), recovery (i.e., body’s capacity and time to return to baseline after strain (a measure incorporating HRV, RHR, sleep performance, respiratory rate and skin temperature)), and training (i.e., calculated as a categorical variable: attending training, off, days shift, night shift).

### 2.5. Focus Groups

Focus groups were conducted with participants from both groups (i.e., firearms officers and mixed staff) to better understand their experiences using the WT. After using the devices, the groups were asked the following questions described in brief below:-What are your thoughts now?-Have your views changed?-Do you understand the data more?-Have you altered any habits?-Have you accessed the support?-Barriers to wearing?-Positives of the band?-Would you recommend the WT?-Could we have done anything differently?-Are there any other aspects to discuss?

### 2.6. Statistical Analysis

Statistical analysis was conducted using IBM SPSS Version 28.0. Analyses were organized in three parts. First, biometric data from the firearms police officers’ group and the mixed police staff group were statistically analyzed to provide insights into each variable and the relationships between variables. The correlation analysis conducted is the Pearson correlation coefficient.

The dataset underwent a thorough analysis called conditional independence graphical analysis [43]. This method systematically examines the statistical relationships between pairs of variables given information about another variable, known as conditional independence. Given the common occurrence of mutual covariance in multivariate analyses, particularly in social sciences, this approach ensures that each association test between variables considers all other variables in the analysis. The resulting structure of multivariate associations is displayed in a map known as a conditional independence map (CI-Map). Pairs of variables lacking significant mutual information are disconnected from the map. “Mutual Information” (MI) measures the strength of association between a pair of variables in comparison to other significant links in the CI-Map. Therefore, utilizing MI offers valuable insights into association strength beyond the mere indication of significance (i.e., *p* < 0.05). It is important to emphasize that these statistical links are associations derived from the data sample, and no causal relationship can be inferred from this analysis.

Second, survey responses were analyzed using descriptive analyses, largely due to small sample sizes prohibiting more advanced statistics. Thirdly, qualitative data from the focus group were analyzed using thematic analysis. The analysis was conducted using Braun and Clarke’s Reflexive Thematic Analysis [44] to establish any patterns across the data being presented. Due to the researcher’s previous experiences working within policing, they used the Semantic Reflexive model to induce themes from the research which allowed them to describe and interpret the data based their experience of UK policing.

## 3. Results

### 3.1. General Wellbeing and Perception of Wellbeing Questions

The results from the general wellbeing and perception of wellbeing questions can be seen in Table 2. In firearms officers, the majority of participants (*n* = 7) either disagreed or strongly disagreed that they had received classroom training about stress and trauma in the context of their work. The mixed police staff group had a more heterogenous response with half of the participants disagreeing (*n* = 14), but ten respondents agreeing (*n* = 10), with a further four indicating a neutral position.

### 3.2. Depression Anxiety Stress Scales (DASS)

The results from both groups completing the DASS scale can be seen in Table 3.

### 3.3. The Perceived Stress Scale (PSS)

The results from the PSS can be seen in Table 4.

### 3.4. The Pittsburg Sleep Quality Index (PSQI)

The results from the PSQI can be seen in Table 5.

### 3.5. The Alcohol Use Disorders Identification Test (AUDIT-C)

The results from the AUDIT-C test can be seen in Table 6.

### 3.6. Biometric Data

Biometric data from the WT can be seen in Table 7.

As Conditional Independence maps (CI-Maps) in Figure 1 and Figure 2 show, much of the variation in the data for both groups can be explained through the individual relationships between HRV and Recovery and between HRV and RHR. Sleep efficiency was not connected to any other variable in the map (coloured green in map). Differently from the firearms officers’ group, The CI-Maps for the mixed staff group (Figure 2) reveal three separate clusters of variables, with any of the sleep-related measures influencing HRV, Recovery, and RHR.

Based on the correlations identified in the CI-Maps above, Scatterplots were conducted to (visually) compare the relationship between HRV, Recovery, and RHR variables among both groups.

The line in Figure 3 shows the weak positive correlation (R = 0.34) between HRV and recovery. Results reveal that the mixed staff group tended to have lower HRV than the firearm officers’ group. The recovery for the mixed police staff group also spans a narrower range of values when compared with the firearm officers’ group.

The line in Figure 4 shows the strong negative correlation (R = −0.72) between HRV and RHR. A low resting heart rate is more likely to indicate that the body is primed to take on the strain (either in the form of stress or physical activity), therefore, has a higher HRV. The mixed staff group has higher RHR and lower HRV than the firearm officers’ group. In the majority, firearms officers display indicators of greater fitness than the control group. Some examples of firearms officers with indicators of reduced fitness more aligned to control group counterparts do exist (Figure 4).

Additional analysis of the data found that there was no noticeable difference between the two study groups with regard to RHR in comparison to average daily strain. There are, however, more points for the mixed police staff group above the trend line than below it. The trend shows that as RHR decreases, strain increases, showing that firearms officers tend to have lower resting heart rates, but with a wider range of daily strain values compared to the mixed police staff group.

### 3.7. Focus Group Responses

Similar themes were found for both groups:

#### 3.7.1. Awareness and Understanding of Data and Wellbeing

Both groups acknowledged an improved understanding of the causes of their mood, stress and wellbeing as a result of being able to see and monitor the data from their devices. Officers from both groups specifically referred to a better understanding of how the shifts impacted their wellbeing. As one firearm officer discussed, the device made him “aware of the impact of shifts and how it affects health”. More specifically, an officer from the mixed group commented on how “night shifts affected [his] body and recovery”. Such improved awareness of their body and wellbeing was also seen as a benefit to demystifying wellbeing concerns: “[sleep patterns] were not as bad as first thought”, and promoting a healthier lifestyle: “really enjoyed it and would recommend it to anyone who is anxious or overthinking/overwhelmed. It helped to keep a work/life balance”.

#### 3.7.2. Management of Their Own Wellbeing

Linked to the better awareness of their body and wellbeing, both groups referred to examples where they took lifestyle changes and monitored their data to improve their general wellbeing. For example, an officer from the mixed group mentioned how the device acted as a motivator to drink less alcohol/caffeine, while for other officers, it promoted increased use of the gym from going “bi-monthly to four to five times a week, “Changed life, started to do more sport”. As a firearm office(s) reflected, “everyone made efforts to improve themselves”.

Participants acknowledge that they were monitoring (through data from their devices) and tracking the impact that changes in their lifestyles had on their wellbeing. As the quotes below demonstrate, most participants from both groups perceived benefits: “changed habits and saw benefits, definitely”, “Better recovery with decreased caffeine and decreased strain and balance”, “running is making my sleep better”.

Linked to the better awareness of their body and wellbeing, both groups referred to examples where they took lifestyle changes and monitored their data to improve their general wellbeing. For example, an officer from the mixed group mentioned how the device acted as a motivator to drink less alcohol/caffeine, while for other officers, it promoted increased use of the gym from going “bi-monthly to four to five times a week”, “Changed life, started to do more sport”. As a firearm office(s) reflected, “everyone made efforts to improve themselves”.

Participants acknowledge that they were monitoring (through data from their devices) and tracking the impact the changes in their lifestyles had on their wellbeing. As the quotes below demonstrate, most participants from both groups perceived benefits: “changed habits and saw benefits, definitely”, “Better recovery with decreased caffeine and decreased strain and balance”, “running is making my sleep better”.

#### 3.7.3. Peer Support

Throughout the conversations, peers were seen as the biggest support. Officers felt their peers promoted talking and comparing their data and their lifestyle: “you can see how you are doing against other age groups and it acts as a motivator”.

#### 3.7.4. Problems with the Device

Despite the benefits of the wearable tech, some problems were noted with the device, which mainly related to (i) inconsistencies with the data—that is, officers noticing their device readings were different from other WT devices—and (ii) the design. Officers from both groups referred to aspects related to battery life, “wearing two devices”, the strap material, and compatibility with personal phone as factors for improvement.

#### 3.7.5. Use of WT within the Force

Finally, there was general agreement among officers from both groups on the benefits of using wearable tech. These included: reducing sickness, a better understanding of the impact that shift patterns have for different ranks, and the opportunities such devices offer to predict trends in biometric data. Yet, there was also emphasis on the need to preserve the anonymity of the data.

## 4. Discussion

### 4.1. Brief Overview of Findings

The results appear to suggest that firearms officers and the police staff group have similar sleep efficiency, but firearms officers have worse sleep consistency and sleep performance. The firearms officers appear to have higher HRV and slightly lower resting heart rate. Both groups spoke positively during the focus groups of how engaging with and utilizing elements of WT such as data monitoring improved their quality of life, giving them a better understanding of sleep, wellbeing and how they had consequently engaged in lifestyle modifications.

### 4.2. Heart Rate Variability

A difference is observed in HRV between firearms officers and the mixed group. This greater HRV might be due to the superior fitness levels required to fulfil these roles compared to non-specialized roles [45]. This greater HRV is likely facilitating considerable recovery [46], as similar recovery is observed between the two groups despite the greater rigors of the firearms role. Better HRV indicates a stronger vagus nerve and vagal tone (nerve activity), this activates the parasympathetic nervous system which will calm inflammatory cardiovascular responses. As such, the physiological expenditure incurred following stress events is reduced and recovery time is shortened [47]. Decreases in HRV are expected acutely during stress activities and individuals possessing a high resting HRV will return to their resting levels and obtain recovery [48]. Individuals with low resting HRV are more likely to see little or no change in their already low HRV during stress activities, with periods of high HRV and recovery also eluding them [46].

### 4.3. Sleep

The firearms officers were evenly split in their perceptions of getting enough sleep and the mixed group of officers predominantly stated they did not get enough sleep. This is similar to their PSQI scores, with increased number of firearms officer participants indicating poor sleep. This sentiment could be explained by the WT which revealed firearms officers had reduced sleep duration, sleep performance and consistency compared to the mixed group.

Given the favorable HRV and RHR seen in firearms officers, their superior cardiovascular profile may be supporting their recovery and sleep efficiency overnight and mitigating against the reduced sleep performance and greater strain observed compared with mixed group officers. Possessing greater fitness will assist in improved recovery during sleep [49]; this greater fitness in firearms officers can be inferred from the improved HRV, RHR in this cohort, and known additional training and job-related standards to complete their roles [45].

The mixed group also had a high prevalence of indicating self-reported poor sleep despite their favorable WT data. The WT data indicate that sleep efficiency was the same and high (90%) for both groups. This suggests that good sleep efficiency alone does not mirror perceived adequate sleep. It is expected that sleep efficiency combined with improved sleep duration, performance, and consistency is required to achieve alignment a subjective improvement in sleep that the individual may categorize as getting adequate sleep.

### 4.4. Wellbeing

This cohort did not display high frequency of severe depression, anxiety, or stress.

These findings are contrary to those typical of these occupational groups in the literature [50]. This may be explained by length of service; Gullon-Scott and Longstaff (2022) [51] found officers were most at risk of experiencing stress in the first 15 years of service, with most of the cohort in the present study being more time-served.

This cohort also predominantly reported a good work–life balance. Work–life balance and wellbeing are established as being strongly connected [52]. Previous research in a police cohort has determined that work–life balance is essential for wellbeing [53] and the present study’s findings could support this. As participation was on a voluntary elective basis, it may be that this study attracted a ‘healthy worker’ population, less likely to be experiencing these mental health difficulties.

There was not wholesale participation in self-care and mindfulness practices in this cohort. The use of mindfulness interventions has been reported as having positive effects on HRV and reductions on psychological strain in police officers [54]. This, therefore, may be a cost-effective intervention to be adopted in this cohort. An array of self-care and mindfulness practices exist including those incorporating PA [55,56]. Given the limited uptake of practices in this cohort and the perception that insufficient physical activity is being obtained it may be indicated that practices related to physical activity would be of benefit. This should further contribute to improved HRV given greater physical activity and fitness levels, further assisting with the management of the demands of this occupation [57]. The firearms officers did report greater exercise despite also reporting perceiving to not receive enough PA. This may be reflective of them completing large amounts of exercise as part of necessary physical training to prepare them for their role [11], but also the long sedentary periods experienced by police populations whilst on stand-by to respond to emergency incidents [58].

### 4.5. Use of WT

Reservations were held by a small number of participants in the Buckingham et al. (2020) [59] study around skin irritation from the device and consequential feelings of guilt and anxiety. Participants in the present study also cited concerns on design and interference of strap material. A further issue raised was with compatibility with their existing devices or having to wear multiple devices, this could present a practical complication for officers involved in active duty due to being involved in physical interactions or use of force with offenders [60]. Inconsistency in readings between devices for participants concurs with previous research [61] and likely created some confusion and skepticism around the reliability of their devices.

The primary concerns in the present were regarding preservation of privacy with respect to data being captured, a feature consistent with other WT studies [34]. Within the present study, the peer support element of the device use was highly regarded, whereas the ‘individual’ phase was preferred to the ‘social’ phase (56% vs. 7%) in the Buckingham et al. (2020) study [59], with participants citing concerns around data privacy in the social phase as one of the reasons for this.

The use of the WT seemingly promoted personal reflection on various health and wellbeing markers. This is encouraging and supports previous research in WT which has been confirmed as having a positive impact on employee wellbeing [30]. Previous research has found acceptance for WT with relation to physical activity levels in police populations [58]. The cohort in Buckingham et al. (2020) [59] returned improved mental health quality of life after wearing the device. When considered with the general acceptance reported in this study and the evaluation of other health and wellbeing markers included, increasingly WT demonstrates a viable practice for adoption for the betterment of health amongst police officers.

At this stage there are many issues a police service would need to consider before implementing the use of WT, but due to the small cohort size within this study, none are significant. A further control study is planned by this research group in 2024 that would give a police service more direction for long-term adoption, but early indications from these studies show the importance of confidentiality and trust of the data and service is key for the long-term success of the program.

### 4.6. Strengths and Limitations

The main strength of this study is the holistic assessment of a multitude of variables in relation to the health and wellbeing of police officers, including firearms officers. The study combines quantitative and qualitative elements to provide more context to the experience of officers utilizing WT. The study provides insight into where future research direction can be applied with this special population and others like it.

This study is primarily limited by its reduced sample size, inhibiting the ability to make significant statistical comparisons between groups, draw significant conclusions about the data, or be considered representative of all police officers. The study sample is also not reflective of true gender representation in police organisations given the low number of females in both groups. The study also did not capture details such as shift patterns, emergency incident exposure, smoking habits, body composition, nutrition, and caffeine consumption. Stress levels will be variable over an officer’s career so greater understanding and exploration of mental wellbeing and its interaction with length of service in relation to the health outcomes is required, which is beyond the scope of this study. These factors could influence the variables measured and future research should, therefore, consider their inclusion for a more comprehensive evaluation.

Given that some participants expressed concerns regarding data privacy and issues with the design of WT devices, these are a limitation to be considered. Addressing these concerns and issues could influence perception and acceptance of WT; therefore, future research should investigate if this encourages more effective implementation of this technology.

## 5. Conclusions

Given the acceptability of WT seen by participants in this study, it indicates a valuable tool which can support both individuals and organizational occupational health and fitness provisions to obtain greater HRV and subsequently sleep and recovery in officers. Thus, preserving components of health and wellbeing are vital in reducing officer absenteeism and attrition. Studies with larger sample sizes and other specific police roles are required to confirm the suitability of WT for widespread use.

## Figures and Tables

**Figure 1 ijerph-21-00186-f001:**
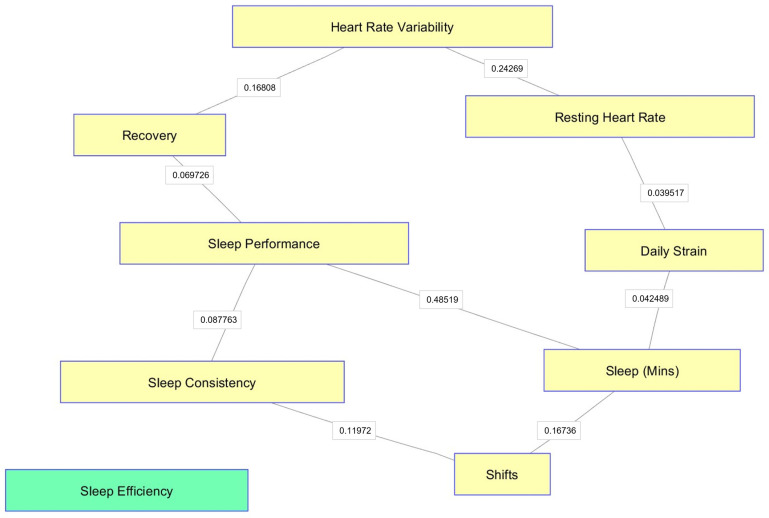
CI-Maps for firearms officers’ group.

**Figure 2 ijerph-21-00186-f002:**
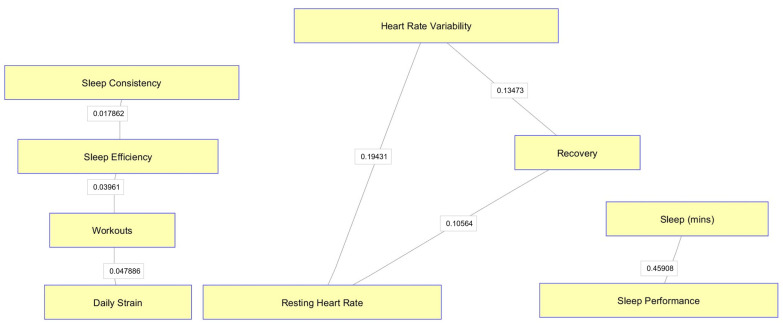
CI-Maps for mixed group.

**Figure 3 ijerph-21-00186-f003:**
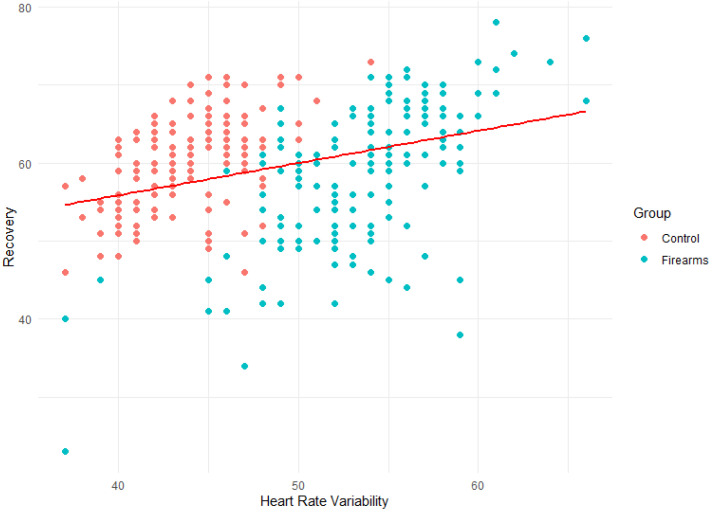
Scatterplot of HRV vs. Recovery.

**Figure 4 ijerph-21-00186-f004:**
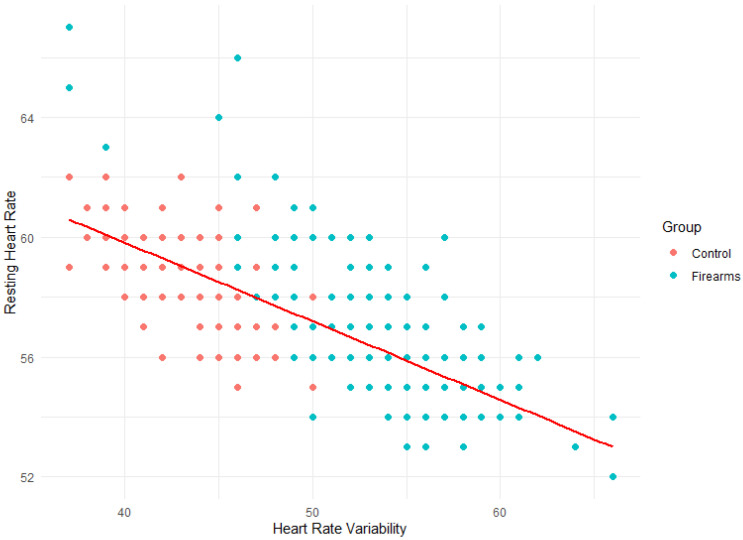
Scatterplot of HRV vs. resting heart rate.

**Table 1 ijerph-21-00186-t001:** Participant descriptive statistics (*N* = 40).

Demographic		Firearms Group % (*n* = 12)	Mixed Group % (*n* = 28)
Gender	Male	100 (12)	85.7 (24)
	Female	-	10.7 (3)
	Prefer not to say	-	3.6 (1)
Ethnicity	White British	100 (12)	96.4 (27)
	White other	-	3.6 (1)
Age	≤35 years	16.7 (2)	10.7 (3)
	36–44 years	50 (6)	53.6 (15)
	≥45 years	33.3 (4)	35.7 (10)
Years of service	0–2	-	7.1 (2)
	3–5	-	-
	6–8	-	3.6 (1)
	9–11	-	3.6 (1)
	12–14	41.7 (5)	28.6 (8)
	15–17	8.3 (1)	10.7 (3)
	18–20	33.3 (4)	14.3 (4)
	21–23	8.3 (1)	7.1 (2)
	24–26	8.3 (1)	10.7 (3)
	27–29	-	14.3 (4)

**Table 2 ijerph-21-00186-t002:** Results from the general wellbeing and perception of wellbeing questionnaire.

Wellbeing Measure	Firearms Group % (*n* = 12)	Mixed Group % (*n* = 28)
Rarely engage in self-care	41.7 (5)	28.6 (8)
Occasionally engage in self-care	16.7 (2)	21.4 (6)
Often engage in self-care	33.3 (4)	28.6 (8)
Routinely engage in self-care	8.3 (1)	21.4 (6)
Never engage in mindfulness	33.3 (4)	39.3 (11)
Rarely engage in mindfulness	33.3 (4)	25 (7)
Occasionally engage in mindfulness	16.7 (2)	17.9 (5)
Often engage in mindfulness	16.7 (2)	10.7 (3)
Routinely engage in mindfulness	-	7.1 (2)
Exercising rarely	8.3 (1)	10.7 (3)
Exercising occasionally	25 (3)	42.9 (12)
Exercising often	33.3 (4)	17.9 (5)
Exercising routinely	33.3 (4)	28.6 (8)
Getting enough physical activity	41.7 (5)	35.7 (10)
Not getting enough physical activity	58.3 (7)	64.3 (18)
Getting enough sleep	50 (6)	28.6 (8)
Not getting enough sleep	50 (6)	71.4 (20)
Have a good work–life balance	91.7 (11)	57.1 (16)
Do not have a good work–life balance	8.3 (1)	39.3 (12)
No response	-	3.6 (1)
Feel overly stressed	25 (3)	42.9 (12)
Do not feel overly stressed	75 (9)	57.1 (16)
Feel anxious	33.3 (4)	28.6 (8)
Rarely feel anxious	58.3 (7)	64.3 (18)
No response	8.3 (1)	7.1 (2)
Do have enough personal time	33.3 (4)	60.7 (17)
Do not have enough personal time	66.7 (8)	39.3 (11)
Have enough time with family/friends	41.7 (5)	60.7 (17)
Do not have enough time with family/friends	58.3 (7)	39.3 (11)

**Table 3 ijerph-21-00186-t003:** DASS scale results.

DASS Depression Interpretation	Firearms Group % (*n =* 12)	Mixed Group % (*n =* 28)
Normal (0–9)	100 (12)	85.7 (24)
Mild (10–13)	-	3.6 (1)
Moderate (14–20)	-	7.1 (2)
Severe (21–27)	-	-
Extremely Severe (28+)	-	3.6 (1)
**DASS Anxiety Interpretation**	**Firearms Group** **% (*n =* 12)**	**Mixed Group** **% (*n =* 28)**
Normal (0–7)	91. (11)	82.1(23)
Mild (8–9)	8.3 (1)	10.7 (3)
Moderate (10–14)	-	3.6 (1)
Severe (15–19)	-	-
Extremely Severe (20+)	-	3.6 (1)
**DASS Stress Interpretation**	**Firearms Group** **% (*n* = 12)**	**Mixed Group** **% (*n* = 28)**
Normal (0–14)	100 (12)	82.1 (23)
Mild (15–18)	-	7.1 (2)
Moderate (19–25)	-	7.1 (2)
Severe (26–33)	-	3.6 (1)
Extremely Severe (34+)	-	-

**Table 4 ijerph-21-00186-t004:** The results from the PSS.

PSS Category	Firearms Group % (*n* = 12)	Mixed Group % (*n* = 28)
Low stress (0–13)	75 (9)	60.7 (17)
Moderate stress (14–26)	25 (3)	32.1 (9)
High stress (27+)	-	7.1 (2)

**Table 5 ijerph-21-00186-t005:** PSQI results.

PSQI Category	Firearms Group % (*n* = 12)	Mixed Group % (*n* = 28)
Good (0–4)	41.7 (5)	14.3 (4)
Poor (5–21)	58.3 (7)	85.7 (24)

**Table 6 ijerph-21-00186-t006:** AUDIT-C results.

Audit-C Classification	Firearms Group % (*n =* 12)	Mixed Group % (*n =* 28)
Low risk (0–4)	41.7 (4)	36 (10)
Moderate risk (5–6)	58 (7)	43 (12)
High risk (7+)	8 (1)	21 (6)

**Table 7 ijerph-21-00186-t007:** Median value of Biometric variables.

Variable	Firearms Group (*n* = 12)	Mixed Group (*n* = 28)
Average sleep (minutes)	395	403
Sleep performance	75	79
Sleep consistency	61	69
Sleep efficiency	90	90
Recovery	61	60
HRV	54	44
Resting heart rate	56	58
Average strain	10.3	9.9

## Data Availability

Restrictions apply to the availability of these data. Data were obtained from police officers and are only available with the permission of the Police Force and the National Police Well Being Service.

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
