# Peer review of "Wearable Technology: A Wellbeing Option for Serving Police Officers and Staff? A Comparison of Results of a Pilot Study with Firearms Officers and a Group of Mixed Officers and Staff"

_ijerph, 2024, doi:10.3390/ijerph21020186_

Round 1

Reviewer 1 Report

Comments and Suggestions for Authors

Thank you for the opportunity to review the paper on the use of wearable technology (WT) in UK police officers. The authors aimed to investigate the acceptability and engagement with WT in order to enhance officer health and wellbeing, an issue which has been the source of previous investigations. The authors reported both quantitative data in the form of demographic information, biometric information derived from a Whoop band, and five self-reported surveys, and qualitative data, elicited from focus groups, which underwent a thematic analysis.

Descriptive statistics, predominately percentages were reported for most of the quantitative data, with some correlations and scatter plots, while qualitative data led to four similar themes identified from both subgroups. The article is well written, the topic does align with the journal, and would be of interest to the reader, however I do have some concerns with the way it is currently presented.

The article is described as a pilot study, aimed at determining acceptability and engagement of WT, which could be achieved with simply the qualitative component of this study. Given that most of the quantitative data is glossed over and not reported extensively in the results, it may be best to split this paper into two separate papers. The current aim fits the qualitative data, while the quantitative data profiles the general well being of UK police officers (albeit a small sub-set). This enables a comparison between the firearms officers and mixed group. There were similar themes between the two groups with respect to the qualitative data, so why separate them? If the data was pooled, if gives you a greater sample size.

Specific comments are below:

Introduction

The introduction reads as being very dot point at times, for example the 2nd, 3rd and 4th sentence are very abrupt and could be joined or expanded upon for a better flow.

Line 39 – extra space at the end of the sentence.

Line 41-42 – Are the 1) and 2) required? Can it not simply state that ‘In a time where crime is becoming more complex… , and there is decreasing public trust… ‘

Line 44 – The authors mention the aim of the study at the end of the first paragraph, prior to setting the scene or providing an extensive rationale. Given that it is also provided on line 107 of the introduction, this mention can be removed.

The first paragraph starts broadly, then focusses on UK police specifically. The second paragraph then speak broadly about well being and PTSD. Perhaps this needs to be restructured to speak broadly for the first few paragraphs, then narrow down on the UK.

The authors allude to ‘firearm officers’. While this might be common nomenclature in the UK, this is not a common term globally. This role should be defined and it should be made clear how this group differs from the ‘mixed group’.

Line 46 – extra spacing after roles.

Line 49-51 – another example of brief sentences, consider combining.

Line 76-79 – Three sentence paragraph? It seems as though both it and the one below are talking about WT, consider joining.

Line 95-95 – which employees were these? Expand on this very brief sentence, or consider joining to the previous.

Line 107 – aim of this study was to (study has been written).

Materials and Methods:

The WHOOP band needs more information. As a minimum, that it is a wrist worn device, manufactured by, enables the user to extract information such as….

There is an extensive amount of rich data in tables 1 through 7 which could be reported on in the results. i.e., is a 100% representation of males as firearm officers common for UK police, 10% female representation seems low for general (mixed) police.

Why these age bins? Why these years of service bins? Were there no officers with 3-5 years of service? Discussion around the fact that firearm officers were of similar age but more experience is warranted, as typically age is a proxy measure for experience. Assuming that firearm officers would be those who are more experienced and have had requisite training to become a firearms officer?

Has the wellbeing questionnaire been used before? Is it valid? Reliable? This outcome measure needs to be better described. What entails the use of self care? Mindfullness? How is physical activity reported? Is it hours/mins/steps? Per day? Per week? Does it include occupational physical activity or only recreational physical activity? How much is deemed to be ‘enough sleep’, is this as perceived by the individual filling out the survey or a set number? How is work-life balance quantified? What one considers to be a healthy work life balance may be different for another.

The DASS, PSS, PSQI, AUDIT-C are well described.

Line 172-179 – no need for vanity caps for Sleep efficiency, Recovery, Heart Rate Variability (although abbreviating to HRV would be warranted here) Workout, Resting Heart Rate, Average strain, Training.

What is the reliability/validity of the measures which the WHOOP band collects? How is this strain calculated? Which HRV variable does the WHOOP band utilise. (it’s RMSDD). There has been extensive work done in the use of HRV in tactical populations, refer:

Tomes et al. (2022) Defining, measuring, and monitoring resilience for the tactical professional: Part 3 – Heart Rate Variability application guide for tactical professionals TSAC Report 65 p4-12.

As part of a three-part series in TSAC report for a table which explains the various uses of HRV and what they are intended to measure. Also see other publications from the same author which may validate the use of HRV for this purpose.

Focus groups – Were these the exact questions asked to participants? i.e. ‘Barriers to wearing”?

Statistical analysis: this section needs much more detail. For part one, for the biometric data, was normality checked? Actioned? Parametric or non-parametric tests? It appears on page 8 and 9 that correlation analyses were conducted, which ones? How should the results be interpreted (moderate, high etc)?

Who conducted the thematic analysis? How was this performed? What sampling strategy was used? which data collection method? Which qualitative approach, paradigm, what were the researcher characteristics and reflexivity?

Results

Reporting on tables 2-7 is required. i.e. firearms officers appear to engage is self care less than the mixed group, appear to exercise more, but report not getting enough physical activity? This seems counter-intuitive.

Similar to the point previously about the survey – what are the limits to the responses? What is ‘rarely’, what is considered to be ‘exercising often’. If this is up to this individual this needs to be stated, however is also a limitation.

The use of the CI maps is interesting, however there was no explanation of the use of them in the statistical analysis section. How should the reader interpret them? What do the colours mean? I can’t find another study in this journal which uses this approach to display data, so the reader will require more information.

Line 238 – was significance determined for these correlations? 0.34 is a low correlation, so is this meaningful? Is it simply confounded by the individuals cardiovascular response?

Results reveal that the mixed staff group tend to have lower HRV… was this compared statistically?

This ‘recovery’ metric which WHOOP reports is based on what? How is this 0-100 calculated?

Were the data for HRV and RHR averaged over the six months for each individual? Consequently isn’t figure 4 common sense? If the WHOOP is exporting RMSDD and values generally increase with cardiovascular fitness, then the individuals who are fitter have lower RHR and lower HRV as measured by RMSDD.

HRV will also be confounded by alcohol, nicotine, caffeine, and shift work. Alcohol use was reported on, but what about the others? Do both groups conduct shift work? It is also individual, there is no comparison to a ‘restful state’ for these individuals.

3.7 onwards – these are the results which achieve your aim of the study and should be the primary focus.

Discussion:

The section titles – main findings – these are not your main findings. Your main findings should align to your studies aim. Was WT accepted and engaged with? Engagement could be quantified by the amount of use. Was there any indication of how many hours per day the WHOOP bands were worn?

Sections 4.1-4.4 are profiling the health and wellbeing of two different groups of police officers and has nothing to do with the studies aim.   

Strengths and limitations

The limited sample size could be increased by pooling both groups for their qualitative data.

Conclusion

The first two sentences are irrelevant to the studies aims, it is more of an incidental finding, which wasn’t the original intent of the study. The conclusion should start from the third sentence.

Author contributions – remove first sentence – this is the instruction from the journal.

References

Extra spacing between ref 1 and 2.

Spacing between ref 8 and 10 is different.

Spacing and font size between 23-27 is inconsistent.

Ref 33 different font size and spacing. 

Reviewer 2 Report

Comments and Suggestions for Authors

Overall, this is an interesting and valuable article. The wellbeing of police officers is a pertinent social issue and one that significantly affects their performance, subsequent public confidence, and legitimacy. As such, the study contributes to the understanding of how leaders and policy makers in the field can begin to better understand the receptivity and impact of wearable technology as a medium for monitoring wellbeing factors and use this data to embed interventions for improving wellbeing. On that basis I feel the article needs to and should be published.

That said, there are some issues that I feel the authors need to address to provide improved clarity to the study, recognize the potential limitations of the associated outcomes, and address the ethical flags identified before it is ready for publication.

These are as follows:

Minor issues:

2.       P2, first major paragraph. I would also include the issue that officers must live under the shroud of investigation for significant periods, research suggests up to 5 years. This plays a high toll on officer wellbeing. Furthermore, the legitimate threat of conviction and imprisonment.

3.       Participants section. Is it possible to outline the rationale for such low numbers of participants as the figure is highly unlikely to be representative of all police officers so what is the reason for this volume i.e. is it a pilot scheme with a view to a more in-depth study for instance? Additionally, this needs to be covered as limitation.

Ethical issues:

4.       Ethics. These are covered at the end of the manuscript, but can the authors add a sentence confirming approval earlier on. The nature of the study sets off some ethical red flags very early (covered next), but reassurance is not received until the end of the manuscript.

5.       Further regarding this issue. I noted that several of the respondents scored extremely high on some concerning rankings related to the DASS, depression and anxiety scales, and the audit-C assessment. I note on page 3, L12, the fact that leaflets and hotline numbers were provided to the officers to address these issues but the leader in me felt this was not adequate, especially for the firearms officers. An officer who is carrying a firearm, potentially depressed and anxious, with an alcohol issue, is a serious and legitimate risk to themselves and the public. Was any additional support provided to these officers? How was disclosure of this to the police service managed, or not? Overall, these are potentially serious questions that should be covered/addressed given the nature of the data and the roles conducted by the participants as it raised some significant concern on my behalf, but the time and detail spent addressing this area was not proportionate to the concerns it raises. As such, some further elaboration is required, or deeper reflection on this finding that is addressed in the discussion or limitations section.

Outcome limitations:

Throughout, I felt that some significant factors were not included in the analysis that would affect the outcomes.

6.       Most specifically this included weight (or BMI), and smoking status. These are key factors that would no doubt affect the bio data outputs but are not included or discussed at any stage. I would either include them, provide rationale why they are not, and/or cover them in the limitations of the paper.

7.       Similarly, the shift patterns of the two groups are not covered. Research already indicates that shift patterns do affect officer health and wellbeing, especially the number and position of rest days, as such, I was left wondering if this also affected the health outputs. This is key as from experience, specialist firearms officers often work less full night shifts than frontline response officers. Furthermore, the pattern in which these are experienced is also often different. These should again be covered or acknowledged as potential limitations that may affect the outcomes identified.

8.       Another factor that is not covered but I was left wondering about (which means other readers may also), is the type of deployments attended by firearms officers vs. the control group. For example, by their nature firearms officers are often reserved for emergency deployment to high-risk incidents with an immediate threat of violence, or firearms specific threats. Whereas the control group will likely attend a vast array of deployments. I would imagine this has an impact on the heart rate variability as attending a HR incident is likely to create a faster and more distinct shift in HRV? Again, can this be covered or acknowledged as a further potential cause or limitation that may affect the outcomes identified?

9.       In the discussion section, it would be interesting to know the authors views on how the use of WT can be normalized beyond the novelty of such a study. I make this observation as I suspect the adherence and short-term impact of the project may be difficult to sustain, as such, what would a police service looking to implement such a program need to consider to ensure long term success?

10.   Limitations section. Given the issues outlined, I feel like the limitations element of the article is too brief. It would be best to flesh out the issues highlighted in this part of the article to acknowledge their potential impact on the outcomes and provide insight on the impact (if any), so readers better understand the overall holistic results, but also so that future studies that may seek to replicate the research can account for these.

I am happy to read any amended version of the article should this be required, and I wish the authors the best of luck on their journey towards publishing this important research.

Comments on the Quality of English Language

There are quite a few minor typographical errors throughout the paper that need to be addressed. Below are some examples but there are others, so a further proofread is required.

a.       P1, L33, the word ‘by’ needs to be input before ‘police officers place’.

b.       P1, L39-40, word officers are used multiple times. I suggest removal of the second usage.

c.       P2, L46, Space between ‘roles  may’ needs removing.

d.       P2, L67, is ‘the’ before Oscar Kilo required?

e.       P3, L11, should be ‘use of the WHOOP band’.

Reviewer 3 Report

Comments and Suggestions for Authors

The introduction could be strengthened by providing a more in-depth discussion of previous research findings. Also, strengthening the theoretical framework, and providing a more in-depth discussion of findings would help situate the current study within the existing literature.

Comments on the Quality of English Language

The English language is clear throughout the manuscript. A few minor grammatical errors could be addressed through light editing. Overall the quality of English is good and the content is easily understandable for academic readers. No major issues are detected that would inhibit comprehension or assessment of the research.

Reviewer 4 Report

Comments and Suggestions for Authors

Dear Authors, concerning your manuscript "Wearable Technology: A Wellness Option for Police Officers and Active Duty Personnel? A Comparison of Results of a Pilot Study with Firearms Officers and a Group of Mixed Officers and Staff", I am sending you some comments:

Title: Splitting the sentence with two colon symbols could unnecessarily interrupt the sentence; it is advisable to use one.

Introduction:

On line 91: Reviews and research on the use of WT have highlighted both the benefits and challenges of using WT to improve health. This can be changed for clarity to read as follows reviews and research on the use of WT have highlighted both the benefits and challenges of using WT for health improvements.

On lines 159 to 161: Typically, the first three questions are administered, and if the respondent scores less than five, they indicate a lower risk of drinking. Scores of five or higher indicate higher-risk drinking behaviors.

On line 174 add Heart Rate Variability (HRV), since this is the first time the term is used in the text and could be abbreviated.

Regarding HRV please elaborate on how it might contribute to better recovery and note any specific implications this has for the cardiovascular health of firearms officers.

On line 304, it may not be necessary to name the brand with connectivity problems unless it appears along with other brands that do not have them. 

In section 4.3 Sleep, the discussion drawn from the data analysis is that "perceptions of sleep do not match actual sleep in this population." The WHOOP band reports a sleep efficiency of 90%; however, sleep quality is not the same as sleep quantity and, ultimately, perceptions about their rest are subjective. Could this be too simplistic a conclusion? And if it is not, consider a bit more elaboration.

An important limitation of this study is the lack of additional data on the participants' overall experience of mental well-being. While noting that this cohort does not show an elevated frequency of depression, anxiety, or stress, no in-depth exploration of how length of service might influence these outcomes has been explored. Previous research suggests that stress levels may vary throughout an officer's career, and the inclusion of this variable could provide a more complete understanding of mental health in this population.

Some participants' expressed reservations regarding data privacy and wearable technology (WT) design issues are also highlighted as a limitation. Addressing these concerns could have influenced the overall perception and acceptance of WT. Future investigations could address these issues more specifically to facilitate more effective implementation of this technology.

Overall, it is a very clear and well explained article, which opens an interesting area to explore in the future. 
